# Relative Importance of Soluble and Microsomal Epoxide Hydrolases for the Hydrolysis of Epoxy-Fatty Acids in Human Tissues

**DOI:** 10.3390/ijms22094993

**Published:** 2021-05-08

**Authors:** Christophe Morisseau, Sean D. Kodani, Shizuo G. Kamita, Jun Yang, Kin Sing Stephen Lee, Bruce D. Hammock

**Affiliations:** Department of Entomology and Nematology, U.C. Davis Comprehensive Cancer Center, University of California Davis, Davis, CA 95616, USA; chmorisseau@ucdavis.edu (C.M.); sean.kodani@joslin.harvard.edu (S.D.K.); sgkamita@ucdavis.edu (S.G.K.); junyang@ucdavis.edu (J.Y.); sing@msu.edu (K.S.S.L.)

**Keywords:** epoxy-fatty acid, epoxide hydrolase, cardiovascular disease, inflammation, pain

## Abstract

Epoxy-fatty acids (EpFAs) are endogenous lipid mediators that have a large breadth of biological activities, including the regulation of blood pressure, inflammation, angiogenesis, and pain perception. For the past 20 years, soluble epoxide hydrolase (sEH) has been recognized as the primary enzyme for degrading EpFAs in vivo. The sEH converts EpFAs to the generally less biologically active 1,2-diols, which are quickly eliminated from the body. Thus, inhibitors of sEH are being developed as potential drug therapeutics for various diseases including neuropathic pain. Recent findings suggest that other epoxide hydrolases (EHs) such as microsomal epoxide hydrolase (mEH) and epoxide hydrolase-3 (EH3) can contribute significantly to the in vivo metabolism of EpFAs. In this study, we used two complementary approaches to probe the relative importance of sEH, mEH, and EH3 in 15 human tissue extracts: hydrolysis of 14,15-EET and 13,14-EDP using selective inhibitors and protein quantification. The sEH hydrolyzed the majority of EpFAs in all of the tissues investigated, mEH hydrolyzed a significant portion of EpFAs in several tissues, whereas no significant role in EpFAs metabolism was observed for EH3. Our findings indicate that residual mEH activity could limit the therapeutic efficacy of sEH inhibition in certain organs.

## 1. Introduction

Poly-unsaturated fatty acids (PUFA) are major structural constituents of cellular membranes, influencing their fluidity and the environment of membrane proteins. Additionally, PUFAs act as a substrate reservoir for the formation of oxidized metabolites, which are bioactive lipid mediators [1,2]. The best understood of these metabolites are those produced by cyclooxygenases and lipoxygenases [3]. However, for the past three decades, cytochrome P450 (CYP450) metabolism has increasingly been seen as important for producing both hydroxy- and epoxy-metabolites which are highly active signaling molecules [4,5]. For epoxide formation, CYP450s effectively oxidize ω-6 fatty acids (arachidonic (ARA) and linoleic (LA) acids) and ω-3 fatty acids (including eicosapentaenoic (EPA) and docosahexaenoic (DHA) acids) at varying rates to produce their respective metabolites, including epoxyoctadeceonic (EpOME), epoxyeicosatrienoic (EpETrE or EET), epoxyeicosatetraenoic (EpETE or EEQ), epoxy-docosapentaenoic (EpDPE or EDP) acids, respectively [6]. This family of natural compounds, called epoxy-fatty acids (EpFA), act as lipid mediators and are involved in numerous biological and physiological systems including but not limited to regulation of vasotension, inflammation, neuroprotection, cardioprotection, pain sensation, angiogenesis and cellular stress [7,8].

Due to the relatively slow production of EpFA from CYP450, regulation of EpFA primarily occurs through their enzymatic hydrolysis by epoxide hydrolases (EH) to their respective 1,2-diol metabolites (also called dihydroxy-fatty acids, DiHFAs) [9]. The major enzyme responsible for this hydrolysis is soluble epoxide hydrolase (sEH, gene name: *Ephx2*), an α/β-fold enzyme that hydrolyzes its substrate through a nucleophilic catalytic triad with water as a co-substrate [10]. Besides EpFAs, sEH metabolizes other natural epoxides such as hepoxilins (HX) or terpene oxides [9]. Blocking sEH through either a chemical or genetic knockout is effective at increasing EpFA and decreasing their respective dihydroxy-metabolites in a number of tissues and disease models [7,8]. Thus, sEH inhibitors have not only been an effective tool for studying and understanding the role of EpFA in disease states but have also been proposed as therapeutics to be used in treating hypertension, pain and other diseases [11].

Although sEH is considered the primary EH responsible for regulating EpFA [12], other EHs may additionally turnover EpFA [13,14]. Microsomal epoxide hydrolase (mEH, gene name *Ephx1*) is primarily considered a detoxication enzyme that hydrolyzes xenobiotic aromatic epoxides such as benzo-[α]-pyrene oxide. In addition to xenobiotic epoxides, this enzyme also turns over endogenous epoxides such as EpFA at lower but still relevant rates [13]. The mEH has comparable k_cat_/K_M_ values to sEH on some substrates, including 8,9-EET and 11,12-EET, but is substantially less active on other substrates, including 14,15-EET and leukotoxin (9,10 EpOME). While sEH inhibition is generally considered more important for regulating EpFA titers, double *Ephx1*/*Ephx2* knockouts are more effective than the single genetic knockout at regulating epoxide/diol ratio on some but not all EpFA regioisomers [14]. In the absence of epoxide hydrolysis, EpFAs are degraded through minor alternative routes that include chain elongation or shortening, hydroxylation and further oxidation [6]. Thus, blocking EH pathways increases EpFAs concentrations, but seldomly more than a few folds. This suggests that, at least for the EpFAs studied so far, they exert their effects over a fairly low and narrow concentration range.

Besides mEH and sEH, two other genes, *Ephx3* and *Ephx4*, are likely to encode EHs based on protein structure homology [15]. EH3, the protein corresponding to Ephx3, is reported to have activity on EpFA comparable to that of sEH [15]. However, genetic knockout of EH3 in mice does not alter concentration or turnover of EpFA in the lung, skin, and stomach; does not alter the inflammatory response to LPS; and generally, shows no observable phenotype [16]. It is possible that this discrepancy is due to relatively higher sEH or mEH concentrations that mask the effect of EH3 deletion. However, EH3 seems to play an important role in the metabolism of linoleate-derived epoxyalcohol and ceramides, which are important for skin barrier function [17,18]. By comparison, no publication to date has recorded catalytic epoxide hydrolase activity for the EH4, the protein corresponding to *Ephx4* [15]. However, EH4 and the *EpHx4* gene have been identified as possible factors in a number of physiological activities. EH4 is one of several factors that is enriched in lipid droplets from sebaceous glands and knockdown of Ephx4 results in the upregulation of sebaceous lipids [19]. Furthermore, *Ephx4* is one of several genes that is modified in the rare appendiceal tumor *Pseudomyxoma peritonei*, but its role in carcinogenesis is poorly understood [20].

To better understand the relative roles and contributions of these four epoxide hydrolases in the metabolism of EpFA, we sought to systematically characterize substrate turnover and preference on recombinant enzymes and tested the relative contributions and abundances of each enzyme in multiple human tissues.

## 2. Results

### 2.1. Recombinant Expression of EH Enzymes

To test and compare the catalytic activity of various EH enzymes, recombinant sEH, mEH, EH3, and EH4 were expressed using baculovirus expression vectors in High-Five insect cells from *Trichloplusia ni*. Preparation and purification of sEH and mEH enzymes were carried out based on standard procedures [21,22,23]. Preparation of EH3 was performed based on a modified procedure from Decker et al. [15]. Interestingly, the EH3 is strongly attached to membranes, and attempts to solubilize it using detergents resulted in total loss of activity; thus, a crude membrane extract was prepared. The EH3 activity was not particularly stable at 4 °C and degraded within days. Thus, the EH3 activity was stabilized with 20% glycerol and frozen at −20 °C until usage. Using a commercial antibody, Western blot analysis suggested that the cloning and expression of EH4 was successful. However, no activity for EH4 was detected under any of the conditions and substrates tested.

### 2.2. Substrate Activity for EH Enzymes

Five radio-labelled (*trans*-stilbene oxide: t-SO; *cis*-stilbene oxide: c-SO; *trans*-diphenylpropene oxide: t-DPPO; *cis*-diphenylpropene oxide: c-DPPO; and juvenile hormone III: JH-III), and two LC/MS/MS-based (14,15-EET and 13,14-EDP) epoxide-containing substrates were screened for use in routine biochemical experiments with EH3 and EH4 [12,24] (Table 1). Consistent with previous observations, [^3^H]-*t*-DPPO has high specific activity and high selectivity for measuring sEH activity (150-fold selectivity ratio) while [^3^H]-*c*-SO showed high specific activity and high selectivity for measuring mEH activity (41-fold selectivity ratio) [23,24]. Additionally, sEH displayed high specific activities on both [^3^H]-*c*-DPPO and [^3^H]-JH-III. Both sEH and mEH were active on 14,15-EET and 13,14-EDP; although, sEH generally had specific activities that were two orders of magnitude greater than that of mEH. EH3 was the most active on [^3^H]-*t*-DPPO and 13,14-EDP but did not have a greater activity on any of the tested substrates relative to sEH or mEH. EH4 did not exhibit any activity on the tested substrates and was not used in any further experiments.

Next, the kinetic constants (K_M_ and k_cat_) were determined to further characterize the catalytic activity of sEH, mEH and EH3 on the endogenous substrates 8,9-EET, 14,15-EET, 13,14-EDP and HXA3 (Table 2). For sEH and mEH, our results are in line with previous publications [12,13,25,26]. However, the activity of EH3 on EpFAs is much lower than initially published [15] and more in line with a recent paper on EH3 activity in the skin [17]. The substrates could be separated into two categories. For 14,15-EET and HXA3, the enzyme specificity (k_cat_/K_M_), which is an approximation of an enzyme velocity at low substrate concentrations ([S] < K_M_), is three orders of magnitude higher on sEH relative to both mEH and EH3. This difference in specificity is due to both an increased K_M_ value and a decreased k_cat_ value. Thus, sEH is likely to be the predominant regulator of 14,15-EET and HXA3 in vivo. Interestingly, it was recently shown that EH3 can hydrolyze epidermal-related epoxy-alcohol fatty acids [17]. However, it is unlikely that EH3 plays a role in the metabolism of the related HXA3 in vivo. 8,9-EET and 13,14-EDP formed a second category for which the enzyme specificity (k_cat_/K_M_) of sEH is only few-fold higher than mEH and two orders of magnitude higher than EH3. This variation is primarily due to differences in the k_cat_ value, while the K_M_ values among the enzymes are comparable. Thus, at low substrate concentrations, such as the ones in vivo [5,10], sEH and mEH are likely to both play a significant role in the in vivo metabolism of 8,9-EET and 13,14-EPD, depending on their relative concentration in tissues and/or selective pharmacological inhibition.

Although these four EpFAs are excellent model substrates for comparing the catalytic efficiency of sEH, mEH, and EH3, they do not necessarily represent the whole range of endogenous substrates for these enzymes. To compare between the multiple substrates, each enzyme was incubated with a solution containing a mixture of the stable regioisomers of EpOME, EET, EEQ, and EDP, each at a [S]_final_ = 1 µM (Figure 1) [27]. 5,6-EET, 5,6-EEQ and 4,5-EDP were excluded since they form cyclized lactones that may confound the observed rates of hydrolysis. The amount of each enzyme used in each assay was adjusted so that the amount of diol formed was in the linear range of the assay. Thus, the size of the bars in Figure 1 is not a direct comparison of an enzyme’s activity for a substrate; rather, the bars represent a relative preference for a particular substrate; the higher the bar the higher the preference. For all three enzymes, 13,14-EDP was the substrate with the highest bar, indicating that the three enzymes have the highest preference for this substrate among the 14 compounds tested. In general, the three enzymes display similar substrate preference profiles with a few important exceptions. For example, the epoxides of linoleic acid (EpOMEs) appear to be good substrates for the sEH with a slight preference for the 12,13-regiosiomer, while they are poor substrates for the mEH and EH3. This suggests that EpOMEs should be preferentially metabolized by sEH in vivo, particularly the 12,13-regioisomer. Consistent with the kinetic data (Table 2), 14,15-EET is preferentially hydrolyzed by sEH. However, the mEH seems to prefer the 8,9- and 11,12-EETs over the 14,15-EET. While the preference pattern for EPAs and DHA epoxides is similar across the three enzymes tested, we did not observe any diol formed by neither mEH nor EH3 for the 17,18-EEQ or 19,20-EDP, suggesting that these substrates are also only hydrolyzed by sEH in vivo. However, in the absence of competition, it is possible that mEH and EH3 hydrolyze 12,13-EpOME, 17,18-EEQ and 19,20-EDP, but at a very low velocity. Interestingly, the terminal epoxides from EPA and DHA (17,18-EEQ and 19,20-EDP, respectively) are preferentially formed by CYP450 when patients are fed an ω-3-rich diet [28]. In addition, the beneficial effects of sEH inhibition are strongly synergistic with EPA and DHA in the diet [29].

### 2.3. Relative Contribution of Epoxide Hydrolases in Tissues

The physiologic contribution of individual EHs in specific tissue types depends on the amount of available substrate, the rate of EH hydrolysis and the relative abundance of individual EHs in the tissue. Although the in vitro kinetic data suggest sEH may be the predominant enzyme responsible for the hydrolysis of EpFA in vivo, mEH and EH3 may be important in tissues or cell types where sEH activity is low and the other EHs are high. To quantify the relative contribution of each enzyme, human tissues S-9 extracts that contain sEH and mEH from the cytosolic and microsomal fractions, respectively [10], were acquired from Xenotech LLC (Kansas City, KS, USA). Additionally, whole tissue extracts were obtained from human tissue samples (BioChain Institute, Newark, CA, USA). Besides the cytosolic and microsomal fractions, these latter extracts also contain the peroxisome fraction where sEH is also found, and the plasma membrane fraction to which the mEH is sometimes attached [10]. In addition, because the subcellular locations of EH3 and EH4 are not known [15], whole cell extracts provide better chances to find these proteins. Finally, because the sEH and EH3 activities are sensitive to the presence of detergent, detergents were not used during extract preparation. Samples purchased as S-9 extracts were pooled from multiple patients and samples purchased as whole cell extracts were prepared from individual patients. To our knowledge, none of the tissue extracts are from the same patient. These samples were collected from a heterogeneous population of individuals (details in Appendix A), which may contribute to the variation in the data, but this was necessary based on experimental constraints.

To determine the concentration of mEH, sEH and EH3 in the tissue extracts, Western blot analysis using specific antibodies and recombinant proteins as standard were used (see Appendix A). A recently developed ELISA against human sEH is 100-fold more sensitive than Western blot-based analysis and matches the *t*-DPPO activity data more closely [30]; however, this ELISA was unavailable to us at the time of our analysis. Interestingly, no EH3 was detected in any of the tissues by Western blot analysis. To obtain more sensitive and accurate determinations of sEH and mEH concentrations in tissues, and in the absence of an ELISA for mEH, *t*-DPPO and *c*-SO were used as selective substrates for sEH and mEH activities, respectively [31] (Table 3). Both sEH and mEH activities were observed in all the tissues tested, yet with great variation among tissues. Based on *t*-DPPO hydrolysis, sEH is highly concentrated in the kidney, liver, intestine, adrenal gland, heart and pancreas. By comparison, based on *c*-SO hydrolysis, mEH concentrations are even higher in numerous tissue extracts, especially in the liver, adrenal gland, lungs (especially in smokers), pancreas and testis. In the heart, a possible target of sEH inhibition [5], the concentrations of sEH and mEH are remarkably similar. Interestingly, there are significant differences in enzyme concentrations between the two kinds of extracts. In the S-9 extracts, the concentrations of mEH are higher in all five tissues tested, especially in the liver, than the concentrations of sEH. In the whole cell extracts of the same four tissues (kidney, liver, lung and intestine), mEH concentrations are still generally higher than sEH but the relative differences are not as pronounced (e.g., mEH concentration in the liver S-9 extract is 20-fold higher than sEH, whereas this difference is only 10-fold in the whole cell liver extract). This suggests that some sEH proteins were not accounted for during the preparation of the S-9 extracts.

To investigate more physiologically relevant substrates, the specific activities of tissue extracts with a relatively large amount of sEH and/or mEH were measured against 5 µM 14,15-EET and 13,14-EDP (Figure 2). These measurements were performed at the physiologic pH of 7.4. This pH is optimal for sEH activity, while mEH activity is at 85% of its optimal pH range (8.0–9.0). The activity on *t*-DPPO (Table 3) correlates better with 14,15-EET (R^2^ = 0.88) and 13,14-EDP (R^2^ = 0.80) than *c*-SO correlates with 14,15-EET (R^2^ = 0.65) and 13,14-EDP (R^2^ = 0.62), even when correcting the *c*-SO data for less than optimal pH conditions. These data indicate that globally, sEH is the primary contributor to EpFA hydrolysis in tissues, but it is not the sole enzyme responsible for it. To quantify the relative importance of each enzyme, the theoretical hydrolysis rates of sEH and mEH in the presence of 5 µM of 14,15-EET or 13,14-EDP were calculated in several tissue extracts (Figure 2). Interestingly, in most tissue extracts, the sum of the theoretical activities is greater than the activity measured, suggesting that at 5 µM, the reaction is not under saturating conditions. The data suggest that for 14,15-EET, sEH is the greater source of hydrolytic activity across all of the tissues tested. For 13,14-EDP, the mEH appears to be the dominant source of hydrolytic activities, especially in the liver and adrenal glands, with the exception of the heart, kidney, intestine, and tongue.

To support this observation, selective inhibitors of sEH and mEH, t-AUCB and 12-HAS, respectively [23,32], were used to probe 14,15-EET and 13,14-EDP hydrolysis activities (Figure 3). For 14,15-EET (Figure 3A), the combination of both inhibitors blocks more than 90% of the hydrolytic activity in all of the tissue extracts tested. The inhibition resulting from the combination of inhibitors is indistinguishable from the sEH inhibitor alone in the kidney and small intestine S-9 extracts as well as in all of the whole cell extracts, except for the adrenal extract. This suggests that sEH accounts for a major part of 14,15-EET hydrolysis in these tissues. The results also show that both sEH and mEH contribute significantly to 14,15-EET hydrolysis for the whole cell adrenal extract and the liver and lung S-9 extracts, especially for smokers, with sEH being a larger (55–80%) contributor to this activity. For the 13,14-EDP (Figure 3B), the pattern of inhibition is globally similar to the one observed for 14,15-EET but with a larger contribution by mEH to the hydrolytic activity, up to 50% in the adrenal and lung extracts.

Physiologically, a mixture of EpFAs with various fatty acid backbones and regioisomers exists inside the cells. To test the possibility that metabolism of a specific EpFA is affected in the tissue extract, the hydrolysis of a cocktail of EpFAs at [S] = 1 µM was measured (Appendix A). The pattern of hydrolysis appears to be globally similar in all of the tissues, with 13,14-EDP being the most metabolized, and represents a mixture of the profiles obtained with the recombinant sEH and mEH. This suggests that these two enzymes account for the totality, or close to, of the EpFA metabolism in tissues.

## 3. Discussion

For the past two decades, epoxy-fatty acids have emerged as important endogenous bioactive regulators of several cellular functions that have implications in numerous diseases [5,7]. These compounds are metabolized in the cells by EHs to the corresponding less biologically active diols [7,9]. The aim of this study was to quantify the relative contributions of mEH, sEH, EH3, and EH4 to the hydrolysis of EpFAs in tissues. Three parameters were tested: hydrolytic activities, substrate selectivity, and selective inhibitors. Out of the four EHs tested, no detectable activity was found for EH4 on any of the tested substrates. EH3 displays some activity for several EpFAs (Table 2), but with a rate much lower than previously reported [15]. In addition, EH3 was not detected in the tissue extracts (Table 3). Taken together, our results indicate that EH3 is not important in the metabolism of EpFAs in human. In genetically modified mice, EH3 was shown to play a role in the regulation of epoxides relevant to skin barrier function [17,18]. The data acquired here support a significant role of both sEH and mEH in EpFAs hydrolysis.

Based on kinetic data, the substrates could be divided into three groups: those that are only metabolized by sEH, such as 12,13-EpOME; those that are significantly hydrolyzed faster by sEH than mEH (k_cat_/K_M_ at least two order of magnitude larger), such as 14,15-EET; and those that have a similar specific rate of hydrolysis (k_cat_/K_M_ within one order of magnitude) for both enzymes, such as 8,9-EET or 13,14-EDP. Besides EpFAs, published data suggest that EET-ethanolamides are part of the last group [33]. For the first two groups, sEH is the major route of EpFAs hydrolysis, while for the third group, both sEH and mEH play a significant part in cellular hydrolysis. Interestingly, epoxy-lipids have different described biology based on the fatty acid backbone and the oxirane regioisomer [5,7,12,13,25]. It is possible that sEH and mEH substrate preference and grouping are related to these biological differences. For example, 12,13-DiHOME and 14,15-EET have specific functions related to sepsis and hypertension, respectively, and are metabolized by sEH. In comparison, 13,14-EpDPE is anti-angiogenic and is metabolized by both sEH and mEH. Use of selective inhibitors in animal models and in human has shown that sEH inhibition results in an increased level of EpFAs, supporting the significant role of this enzyme for EpFAs metabolism in vivo [5,7,34,35]. No inhibitor sufficiently effective in vivo exists for mEH yet. However, recombinant animals in which both sEH and mEH are knocked out displayed a much higher level of 14,15-EET than the ones with only one EH gene eliminated, supporting the role of both enzymes for metabolism of this EpFA [14]. The fact that the sEH KO mice have much higher levels of 14,15-EET than the mEH KO mice supports our observation that sEH plays a much higher role in this EpFA metabolism than mEH.

While it is assumed that blood levels roughly mirror the role of EpFAs and each EH at the whole organism level, this could hide the intricate and/or specific function of the EpFAs and EHs in a tissue or organ. Whether a particular biological function is dominated by locally produced or systemic levels of EpFAs is a very open question. Likely, the relative importance of local or systemically produced EpFAs probably varies with the tissue, the biological process and the EpFA. The concentrations of sEH and mEH in human organ extracts are highly variable, ranging several orders of magnitude (Table 3). While in the liver, the sEH and mEH appear generally distributed, in numerous organs, the enzymes are highly localized in specific tissues or cells [36,37,38]. Even in the liver, there are differences between lobular and portal regions and specific cell types. Thus, some individual cells are likely to have concentrations of these enzymes that are higher than the one calculated herein. As the concentration of EpFAs in human is in the high picomolar to low nanomolar range [39], the concentrations of either enzyme are higher than the concentrations of their natural substrates. In such conditions, a large part of both enzymes are idle at any instant. Consequently, a reduction in metabolism through inhibition will happen only if most of the enzyme is inhibited. This relative concentration of enzymes and substrates likely explains why slow, off-rate thigh binding inhibitors of sEH are so effective in vivo [40]. In addition, the metabolism of the substrates is proportional to the specific constant (k_cat_/K_M_) and the ability of the enzyme to access and bind the substrate. At the intracellular level, the mEH is associated with the endoplasmic reticulum (ER) where it is located in proximity of CYPs that generate EpFAs. [6]. While mEH can directly metabolize newly produced EpFAs [14], a large quantity of them are still stored in the cell membranes [6]. It is not known if the EpFAs pool varies among cellular membranes. Nevertheless, when a cell is activated and EpFAs are released into the cytosol for biological activity, the EpFAs become more accessible to the sEH that is free of restriction, resulting from its association with the ER membrane or other membranes, unlike the mEH. The sEH is present in both the cytosol and peroxisome lumen [36] where the concentrations and types of EpFAs available could be different, leading to different biological activity [5,7].

There are clear limitations to this study, which relies heavily on enzymes assays and kinetic studies with pure or enriched proteins in diluted buffered solutions. In contrast, in the cells, the enzymes are in highly concentrated solutions of themselves and other proteins that are probably arranged in complex and dynamic scaffolds, about which we know little. Obviously, even the gross structure of the cells is disrupted by homogenization, thus perturbing natural segregation of the proteins and substrates inside the cells. A limitation of our approach of using whole tissue and S9 homogenates is that it cannot capture the highly localized nature of enzymatic function. The expression and function of sEH and mEH are known to be cell-type specific within an organism. For example, knocking out sEH expression specifically in podocytes is sufficient for reducing diabetic injury in the whole kidney [41]. Compartmentalization within specific organelles may similarly be important to EH function. For example, the peroxisomal sEH has been suggested to have a different role than the cytosolic sEH [42]. Future experiments on cultured cells or whole isolated tissues, where cell and tissue structure are preserved, may serve as a bridge between the in vitro observations in tissue homogenates and translation to human subjects. Separately, while efforts were made to estimate how the enzymes interact with multiple substrates presented at the same time (Figure 1 and Appendix A), the intracellular concentrations of each substrate are not known. In addition, this work focuses on EpFAs as substrates, but other natural epoxides could also be substrates for the studied enzymes and present in the cells at the same time. For example, sEH was reported to also hydrolyze terpene epoxides, while mEH was reported to hydrolyze steroid epoxides [9,10], thus complicating the extrapolation of our results to the in vivo reality. The mEH also plays an important role in the metabolism of epoxide-derived xenobiotics, such as styrene or naphthalene [13]. Put together, in vivo, both endogenous and exogenous alternative substrates likely compete with the EpFAs for sEH and mEH hydrolytic activities. While not perfect, the studies described here hopefully complement in vivo studies in animal models and provide some data for translation to human organs. Emerging technologies will hopefully allow more accurate cell-specific measurement of these enzymes with a smaller sample size. Such technologies will allow the association of sEH, mEH and EpFAs in human to be tested with endogenous factors, such as sex and age, and physiopathological conditions across many organs.

In conclusion, our results show that in human tissues, sEH and to a lesser degree mEH are important in the metabolism of biologically active EpFAs, while EH3 and EH4 do not appear to play a role. The relative importance of both sEH and mEH to a particular EpFA and tissue will most likely be debated over the next decade.

## 4. Materials and Methods

### 4.1. Materials

The EpFAs and radioactive substrates used herein were previously synthesized in the laboratory [12,31]. Other chemicals and biochemical were obtained from commercial sources and used without further purification nor modification. The human tissue extracts were obtained from either Xenotech LLC (Kansas City, KS, USA) or BioChain Institute (Newark, CA, USA). The samples were kept at −80 °C until usage.

### 4.2. Generation of Recombinant Ac-hEH3 and Ac-hEH4

Recombinant baculoviruses Ac-hEH3 and Ac-hEH4 that express the human epoxide hydrolases EH3 (also known as EPHX3 and ABHD9) and EH4 (also known as EPHX4 and ABHD7), respectively, were generated by standard procedures, as described previously [43]. In brief, Ac-hEH3 and Ac-hEH4 were generated following the transfection of Sf-9 cells (Invitrogen, Carlsbad, CA, USA) with Bsu36I-digested BacPAK6 baculovirus DNA (Clontech, Mountain View, CA, USA) and the recombinant transfer vector pAcUW21-hEH3 (see below) or pAcUW21-hEH4 (see below), respectively, using Cellfectin II Reagent (Invitrogen, Carlsbad, CA, USA) under conditions recommended by the manufacturer. Following the transfections, Ac-hEH3 and Ac-hEH4 were isolated from the supernatant of the transfected Sf-9 cells by three rounds of plaque purification on Sf-9 cells. The Sf-9 cells were cultured at 27 °C on ExCell 420 medium (SAFC Biosciences, Lenexa, KS, USA) that was supplemented with fetal bovine serum (2.5% *v*:*v*).

In order to generate pAcUW21-hEH3, a full-length sequence of human EH3 (Appendix A) was PCR-amplified from a template plasmid obtained from the Integrated Molecular Analysis of Genomes and their Expression (I.M.A.G.E.) Consortium (Clone ID 40146982) through Open Biosystems. PCR reactions were performed using KOD Hot Start polymerase (EMD Millipore) with primers EH3BglATG (5′-GAAGATCTATGCCGGAGCTGGTGGTGACCG-3′) and EH3stopEco (5′-CGGAATTCCTAGTCCAGCAGGTCTTGCAAGAAGGC-3′) following the manufacturer’s recommended procedures. The use of primers EH3BglATG and EH3stopEco placed BglII and EcoRI sites (underlined nucleotides in the primer sequences above) at the 5′- and 3′-ends of the coding sequence, respectively. The amplicon (1.1 kbp) was inserted into the cloning vector pCR-Blunt II-TOPO (Invitrogen) and sequenced in both directions in order to confirm the authenticity of the sequence. Subsequently, the 1.1 kbp-long insert was excised by digestion with BglII and EcoRI, and then ligated using T4 DNA ligase (New England BioLabs, Ipswich, MA, USA) into the BglII and EcoRI sites of the baculovirus transfer vector pAcUW21 [43] in order to generate pAcUW21-hEH3. The recombinant transfer vector pAcUW21-hEH4 carrying the human EPHX4 coding sequence (Appendix A) was generated using a similar strategy with primers EH4BglATG (5′-GAAGATCTATGGCGAGGCTGCGGGATTGC-3′) and EH4stopBgl (5′-GAAGATCTTCAATCTTTTTTTCTTGTTTCTTCTTTTAG-3′) and template DNA (Clone ID 5244314) from the I.M.A.G.E. Consortium. The use of primers EH4BglATG and EH4stopBgl placed BglII sites (underlined nucleotides in the primer sequences) at both ends of the coding sequence. The 1.1 kbp-long amplicon was inserted into pCR-Blunt II-TOPO, sequenced in both directions, and ligated into the BglII site of pAcUW21, as described above, in order to generate pAcUW21-hEH4.

### 4.3. Protein Production and Purification

Recombinant sEH and mEH were produced and purified as described [12,23]. Recombinant EH3 and EH4 were produced in High Five cells (Invitrogen) that were grown at 27 °C in ESF921 medium (Expression Systems, Davis, CA, USA). Commonly, the High Five cells (1 × 10^6^ cells mL^−1^) were inoculated with Ac-hEH3 or Ac-hEH4 at a multiplicity of infection of 0.5 and collected at 65 h post inoculation by centrifugation (500× *g*, 20 min, 5 °C). Cell pellets were frozen at −80 °C until usage. The cells were suspended in chilled sodium phosphate buffer (20 mM, pH 7.4) containing 1 mM of DTT, EDTA and PMSF and then homogenized with a Polytron (Brinkman, Westbury, NY, USA) equipped with a 1 cm wide probe (9000 rpm for 30 s, 1 min on ice and 9000 rpm for 30 s). The mixture was centrifuged at 100,000× *g* for 60 min at 4 °C. The pellet was resuspended in sodium phosphate buffer (0.1 M pH 7.4) containing 20% glycerol. The mixture was aliquoted and kept at −20 °C. Solubilization of the targeted protein with any detergent (e.g., 0.1% of Triton X-100, Tween-20, Tergitol NP-40, sodium deoxycholate or lubrol PX) resulted in a total loss of activity. Protein concentration was quantified using the Pierce BCA assay (Pierce, Rockford, IL, USA), using Fraction V bovine serum albumin (BSA) as the calibrating standard.

### 4.4. Enzyme Activities and Inhibition

Enzyme activity using radio-labelled substrates was measured as previously described [24,31]. Kinetic parameters for a series of mono epoxy-fatty acids were determined under steady-state conditions using recombinant human sEH, mEH and EH3 as described [12]. One microliter of substrate solution in ethanol ([S]_final_ from 1.0 to 50 μM; 7 to 8 concentrations used for each substrate) was added to 100 μL of enzyme solution in sodium phosphate buffer (0.1 M pH 7.4) for the sEH and EH3, or Tris/HCl buffer (0.1 M pH 9.0) for mEH, containing 0.1 mg/mL of lipid free BSA. The reaction mixtures were incubated at 37 °C for 5 to 15 min. The reactions were then quenched by adding 400 μL of methanol, and 1 μL of internal standard (9,10-dihyoxy-heptadecanoic acid). The quantity of diol formed was then determined by LC/MS-MS analysis as described [12]. Kinetic parameters were determined by non-linear regression of the Michaelis–Menten equation using SigmaPlot (v14.0, Systat Software Inc., Chicago, IL, USA). An F-test and *t*-test (*p* < 0.05) were performed to ensure the quality of the fitting. Enzyme preference pattern was determined by exposing the enzymes to a mixture of 14 EpFAs each at a final concentration of 1 µM as described [27].

Enzyme activities in tissues were measured with either radioactive substrates or EpFAs following previously described procedures [12,24,31]. The effects of selective inhibitors were measured with 5 µM of 14,15-EET or 13,14-EDP as substrates as described [12] in the presence of 10 µM of either a sEH-specific inhibitor (*trans*-4-[4-(3-adamantan-1-yl-ureido)-cyclohexyloxy]-benzoic acid; t-AUCB [32]) or a mEH selective inhibitor (12-hydroxy-stearamide; 12-HAS 10 µM [23]) or a mixture of both. Controls were run using the radioactive substrates, [^3^H]-*t*-DPPO and [^3^H]-*c*-SO selective for sEH and mEH [31], respectively, confirming that at 10 µM, the inhibitors completely and selectively inhibit their target enzyme (sEH for t-AUCB and mEH for 12-HAS) [23,32].

### 4.5. Western Blot Analysis

Protein samples were diluted in 4X LDS Sample Buffer and 10X Reducing Buffer (Life Technologies, Carlsbad, CA, USA) to provide the given concentrations. Samples were then loaded on Bolt 4–12% Bis-Tris gels (Life Technologies) and run at 200 V for 35 min. Samples were transferred to nitrocellulose membranes and total protein loading was determined by Ponceau S staining. Samples were blocked with 3% BSA and incubated with either primary rabbit polyclonal anti-human sEH antibody (1:5000) (Yu et al. 2004), rabbit polyclonal anti-human EH1 antibody (1:5000) (Thermo Scientific, Waltham, MA, USA, Lot # QD2009572) or rabbit polyclonal anti-human EH3 antibody (1:5000) (MyBioSource, San Diego, CA, USA, Lot# MBS851755) and subsequently incubated with secondary goat anti-rabbit IgG-HRP (1:5000) (Abcam, Cambridge, MA, USA). Membranes were developed with SuperSignal West Femto ECL Detection Reagent (Thermo Fisher Scientific, Waltham, MA, USA) and imaged with ChemiDoc MP (Bio-Rad Laboratories, Hercules, CA, USA). Quantification was performed from quantifying band intensity using ImageLab 5.0 (Bio-Rad). Values are given as the average ± standard deviation of at least 3 independent gels.

### 4.6. Statistics

Results are presented as average ± standard deviation unless noted otherwise. An F-test and *t*-test (*p* < 0.05) were performed to determine statistical differences.

## Figures and Tables

**Figure 1 ijms-22-04993-f001:**
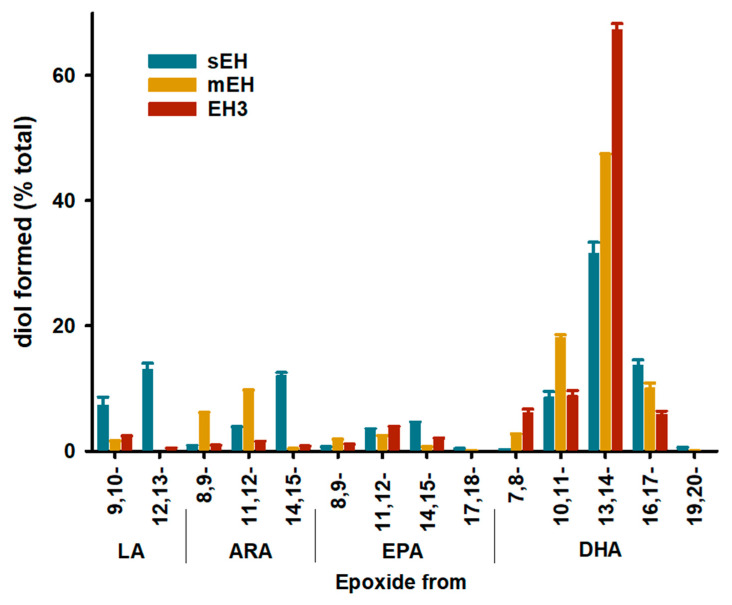
Substrate selectivity of the recombinant human EHs. The enzymes (human sEH 0.09 µg/mL; human mEH 10 µg/mL; human EH3 160 µg/mL) were incubated with a mixture of 14 EpFAs, each at a final concentration of 1 µM for 15 min at 37 °C in sodium phosphate buffer (0.1 M pH 7.4) for sEH and EH3, or Tris-HCl buffer (0.1 M pH 9.0) for mEH, both buffers containing 0.1 mg/mL of BSA. The amounts of diol produced were quantified by LC/MS-MS [27]. Results are average ± standard deviation (*n* = 3). LA: linoleic acid, ARA: arachidonic acid, EPA: eicosapentaenoic acid, DHA: docosahexaenoic acid.

**Figure 2 ijms-22-04993-f002:**
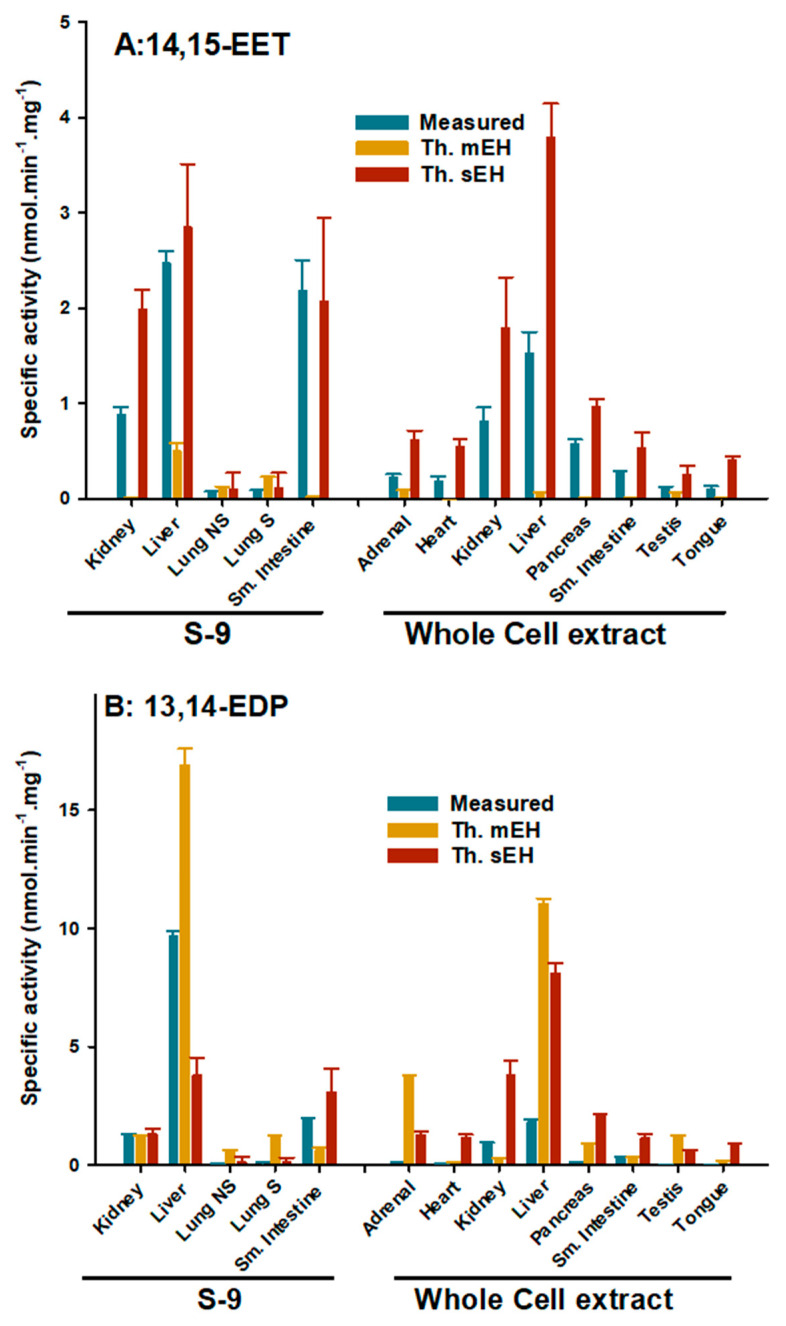
Comparison of human tissue extracts hydrolysis of 5 µM of 14,15-EET (**A**) and 13,14-EDP (**B**), and theoretical activity of sEH and mEH based on the enzyme concentrations (Table 3) and enzyme specificity (k_cat_/K_M_) (Table 2). The 14,15-EET and 13,14-EDP activities were measured in sodium phosphate buffer (0.1 M pH 7.4) containing 0.1 mg/mL of BSA, which is closer to physiological pH.

**Figure 3 ijms-22-04993-f003:**
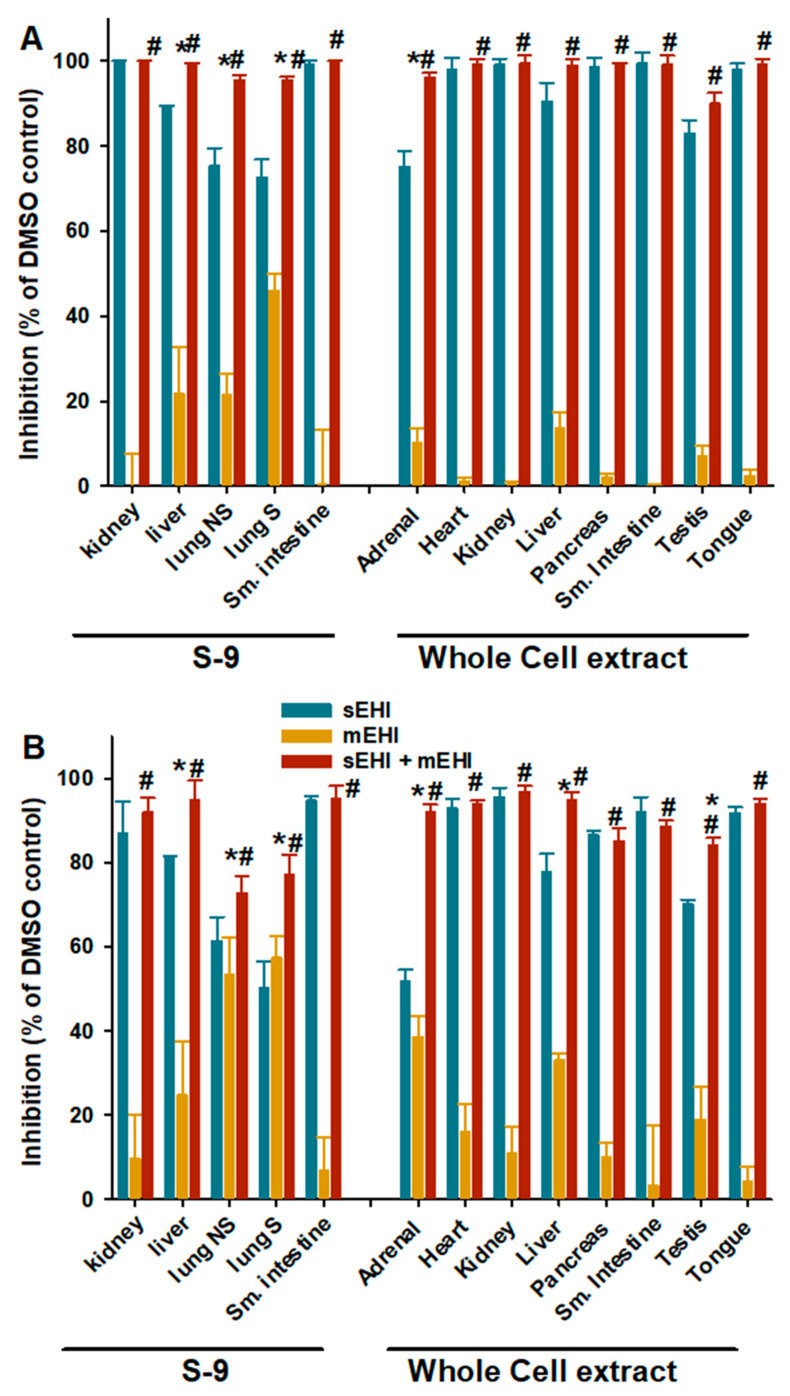
Effect of selective inhibition of sEH (t-AUCB 10 µM) or mEH (12-HAS 10 µM) or both enzymes (mixture of both inhibitors at 10 µM each) in tissue extracts on the hydrolysis of 5 µM of 14,15-EET (**A**) or 13,14-EDP (**B**). The 14,15-EET and 13,14-EDP activities were measured in sodium phosphate buffer (0.1 M pH 7.4) containing 0.1 mg/mL of BSA, which is closer to physiological pH. * Mixture of inhibitors is significantly different from sEH inhibitor alone (*p* < 0.05); # mixture of inhibitors significantly different from mEH inhibitor alone (*p* < 0.05).

**Table 1 ijms-22-04993-t001:** Specific activity of recombinant human EHs for surrogate and natural substrates.

Substrate	Specific Activity (nmol·min^−1^·mg^−1^)			
	sEH	mEH	EH3	EH4
[^3^H]-*t*-SO	90 ± 5	2.0 ± 0.3	1.1 ± 0.2	<0.1
[^3^H]-*c*-SO	12 ± 1	490 ± 20	0.3 ± 0.1	<0.1
[^3^H]-*t*-DPPO	7100 ± 200	1.3 ± 0.3	48 ± 5	<0.1
[^3^H]-*c*-DPPO	1200 ± 200	0.7 ± 0.3	<0.1	<0.1
[^3^H]-JH-III	440 ± 30	<0.1	3 ± 1	<0.1
14,15-EET	1100 ± 25	8 ± 1	19 ± 2	<0.1
13,14-EDP	3300 ± 50	54 ± 5	94 ± 4	<0.1

Results are average ± standard deviation (*n* = 3). Activities were measured at 37 °C in sodium phosphate buffer (0.1 M pH 7.4) for sEH, EH3 and EH4, or Tris-HCl buffer (0.1 M pH 9.0) for mEH, both buffers containing 0.1 mg/mL of BSA. For all substrates, [S]_final_ was 50 μM. The amount of protein and duration of incubation were chosen to result in less than 10% conversion of substrate.

**Table 2 ijms-22-04993-t002:** Kinetic constants of recombinant human EHs for four natural substrates.

Enzyme	K_M_ (μM)	V_m_ (nmol·min^−1^·mg^−1^)	r^2^	k_cat_ (s^−1^)	k_cat_/K_M_ (s^−1^·μM^−1^)
	**8,9-EET**				
sEH	26 ± 1	535 ± 30	0.97	0.56 ± 0.03	0.022 ± 0.001
mEH	2.5 ± 0.6	38 ± 3	0.99	0.032 ± 0.002	0.013 ± 0.001
EH3	10 ± 0.5	44 ± 1	0.99	0.030 ± 0.001	0.003 ± 0.0002
	**14,15-EET**				
sEH	7.0 ± 0.3	2900 ± 100	0.95	3.0 ± 0.1	0.43 ± 0.01
mEH	80 ± 6	53 ± 3	0.99	0.044 ± 0.003	0.00055 ± 0.00001
EH3	76 ± 16	27 ± 4	0.99	0.018 ± 0.003	0.00024 ± 0.00001
	**13,14-EDP**				
sEH	3.1 ± 0.3	2800 ± 100	0.96	2.85 ± 0.09	0.92 ± 0.07
mEH	2.9 ± 0.4	445 ± 14	0.99	0.37 ± 0.01	0.13 ± 0.02
EH3	4.3 ± 1.2	121 ± 9	0.95	0.08 ± 0.01	0.019 ± 0.003
	**HXA3**				
sEH	59 ± 9	7000 ± 670	0.99	7.3 ± 0.7	0.12 ± 0.01
mEH	51 ± 14	23 ± 4	0.98	0.019 ± 0.003	0.00037 ± 0.00006
EH3	14 ± 1	17 ± 1	0.99	0.014 ± 0.001	0.00101 ± 0.00003

Results are average ± standard deviation (*n* = 3). EH3 values were calculated with purity estimated at 5% by SDS-PAGE. mEH values were calculated with purity estimated at 80% by SDS-PAGE. The sEH data are from Ref [12,25,26]. EET: epoxy-eisosatrienoic acid; EDP: epoxy-docosapentaenoic acid; HXA3: hepoxilin A3.

**Table 3 ijms-22-04993-t003:** sEH and mEH activity and concentration in human tissue extracts.

Extract Type	Tissue	[Protein] (mg/mL)	Specific Activity (nmol·min^−1^·mg^−1^)		[sEH] (nM)	[mEH] (nM)
			[^3^H]-*t*-DPPO	[^3^H]-*c*-SO		
S-9	kidney	4.0	3.7 ± 0.3	0.88 ± 0.05	62 ± 12	130 ± 50
	liver	20.0	8.5 ± 1.8	13.5 ± 1.2	442 ± 10	8,700 ± 900
	lung non-smoker	5.0	0.21 ± 0.02	0.36 ± 0.05	4.2 ± 0.4	72 ± 9
	lung smoker	5.0	0.22 ± 0.02	0.85 ± 0.02	4.2 ± 0.4	153 ± 4
	intestine	5.0	6.5 ± 0.7	0.42 ± 0.03	80 ± 5	82 ± 6
Whole cell extract	Adrenal	3.0	2.1 ± 0.2	2.7 ± 0.5	14 ± 2	290 ± 50
	Esophagus	5.0	0.41 ± 0.03	0.07 ± 0.01	4.7 ± 0.4	12 ± 1
	Heart	3.0	1.9 ± 0.1	0.09 ± 0.04	13 ± 1	10 ± 4
	Hippocampus	3.0	0.71 ± 0.05	0.16 ± 0.02	4.8 ± 0.4	17 ± 2
	Kidney	3.0	6.2 ± 0.2	0.2 ± 0.1	42 ± 1	22 ± 11
	Liver	3.0	13.1 ± 0.4	7.9 ± 0.5	88 ± 3	850 ± 50
	Lung	3.0	0.27 ± 0.06	0.19 ± 0.02	1.8 ± 0.4	20 ± 2
	Ovary	3.0	0.17 ± 0.03	0.05 ± 0.01	1.2 ± 0.2	6 ± 1
	Pancreas	3.0	3.3 ± 0.5	0.62 ± 0.03	22 ± 3	67 ± 4
	Skin	2.67	0.6 ± 0.2	0.06 ± 0.02	3 ± 1	6 ± 2
	Small Intestine	3.0	1.8 ± 0.1	0.22 ± 0.02	12 ± 1	24 ± 2
	Spleen	3.0	0.5 ± 0.1	0.05 ± 0.01	3 ± 1	6 ± 1
	Stomach	5.0	0.14 ± 0.04	0.07 ± 0.01	1.6 ± 0.5	12 ± 2
	Testis	3.0	0.9 ± 0.1	0.89 ± 0.07	5.8 ± 0.4	96 ± 7
	Tongue	5.0	0.9 ± 0.1	0.07 ± 0.01	10 ± 1	12 ± 1

Results are average ± standard deviation (*n* = 3). *t*-DPPO activity was measured in sodium phosphate buffer (0.1 M pH 7.4) and *c*-SO activity was measured in Tris-HCl buffer (0.1 M pH 9.0), both buffers containing 0.1 mg/mL of BSA, which are the optimal pH conditions for sEH and mEH, respectively. The sEH and mEH concentrations were calculated based on the specific activity of the purified recombinant enzymes (Table 1).

## Data Availability

Data is contained within the article or Appendix A.

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
