# Peer review of "Relative Importance of Soluble and Microsomal Epoxide Hydrolases for the Hydrolysis of Epoxy-Fatty Acids in Human Tissues"

_ijms, 2021, doi:10.3390/ijms22094993_

Round 1

Reviewer 1 Report

Dear Authors

The manusript provides some new data on epoxide hydrolases and is so of interest for the specialist but also for some general readers.

Some minor comments to improve:

  • check italics style of latin words
  • Fig2 is a bit small; hence hart to read
  • accession numbers and genes used can be placed in suppl mat
  • chapter 4.3; protein expression is wrong; only a gene can be expressed. a protein is produced.
  • rpm is not clear; u need to add the rotor otherwise one cannot reproduce it

The supplemental matieral is well prepared and should be published along.

Author Response

Comments and Suggestions for Authors

Dear Authors

The manusript provides some new data on epoxide hydrolases and is so of interest for the specialist but also for some general readers.

Some minor comments to improve:

check italics style of latin words.

Response: we carefully checked the manuscript to ensure that all the Latin words were italized.

Fig2 is a bit small; hence hart to read

Response: following the reviewer recommendation, the 2 panels of figure 2 were moved above each other allowing larger size for each panel. A similar action was also applied to figure 3.

accession numbers and genes used can be placed in suppl mat

Response: Following the reviewer recommendation, a new table (S4) was added in the supplemental material document where accession numbers and genes used information are available.

chapter 4.3; protein expression is wrong; only a gene can be expressed. a protein is produced.

Response: following the reviewer recommendation, the title of section 4.3 was changed to “Protein production and purification”.

rpm is not clear; u need to add the rotor otherwise one cannot reproduce it

Response: In section 4.3, the instruments characteristics were added.

The supplemental material is well prepared and should be published along.

Response: Thank you

Reviewer 2 Report

Morisseau and coauthors have investigated the Relative importance of soluble and microsomal epoxide hydrolases for the hydrolysis of epoxy-fatty acids in human tissues. Their results show that in human tissues sEH and to lesser degree mEH are important in the metabolism of biologically active epoxy-fatty acids (EpFAs). The authors also reported that their data support a significant role of both sEH and mEH in EpFAs hydrolysis.

This is one excellent study on an important topic. In general, the manuscript well prepared and the study carefully designed and mapped out. Data is convincing, well-presented and discussed. This study brings very important knowledge regarding the importance of sEH and mEH for the hydrolysis of EpFAs in human tissues. These results may trace pathways for better understanding the relative roles and contributions of epoxide hydrolases to the metabolism of EpFAs.

The sample size is only 3, which is too small to get convincing results. However, the authors reported 15 samples from 15 donors, can the authors include more samples in their study?

Authors should differentiate young adult from aged samples. Is there any gender or age differences?

Author Response

Comments and Suggestions for Authors

Morisseau and coauthors have investigated the Relative importance of soluble and microsomal epoxide hydrolases for the hydrolysis of epoxy-fatty acids in human tissues. Their results show that in human tissues sEH and to lesser degree mEH are important in the metabolism of biologically active epoxy-fatty acids (EpFAs). The authors also reported that their data support a significant role of both sEH and mEH in EpFAs hydrolysis.

This is one excellent study on an important topic. In general, the manuscript well prepared and the study carefully designed and mapped out. Data is convincing, well-presented and discussed. This study brings very important knowledge regarding the importance of sEH and mEH for the hydrolysis of EpFAs in human tissues. These results may trace pathways for better understanding the relative roles and contributions of epoxide hydrolases to the metabolism of EpFAs.

The sample size is only 3, which is too small to get convincing results. However, the authors reported 15 samples from 15 donors, can the authors include more samples in their study? Authors should differentiate young adult from aged samples. Is there any gender or age differences?

Response: We understand the reviewer point; however, such study is beyond the scope of this paper as it will require the analysis of samples from hundreds of subject to account for natural variations and underlying conditions. Nevertheless, on line 388, the following statement was added: “Emerging technologies hopefully will allow more accurate cell specific measurement of these enzymes with smaller sample size. Such technologies will allow to test in human the association of sEH, mEH and EpFAs with endogenous factors, such as sex and age, and physio-pathological conditions across many organs.”

Reviewer 3 Report

This is an in vitro study of four epoxide hydrolase enzyme systems which involved measurement of hydrolytic activity, substrate selectivity and pharmacologic selective inhibition. Study used both test systems of purified aqueous enzyme, and their quantitation in 15 human tissue homogenates. Goal was to characterize and quantitate activity of soluble epoxide hydrolase (sEH), microsomal epoxide hydrolase (mEH), epoxide hydrolase 3 (EH3) and epoxide hydrolase 4 (EH4). The overarching purpose was to assess potential theoretical contribution of each of these four EH systems to turnover of epoxy-fatty acids (EpFAs) in human tissues.  Human tissue extracts were either obtained from single patient providing broad spectrum of tissues or from pooled specimens (S-9) derived from multiple patients, both with unknown comorbidity.

Results show that sEH has favorable biochemical enzymatic characteristics for physiological clearance of EpFAs, and that mEH also could play a role in some tissues.  Potential backup role of MEH is supported by its much larger concentration (>10-fold) than sEH in liver, lung and adrenal tissue in particular. MEH was also substantially higher (2-10x) concentration than sEH in kidney, hippocampus, pancreas and stomach.  Other regions of brain were not reported.  Authors did not report whether there was any association of tissue EH content among different tissues derived from same patient. Physiological roles of EH3 and EH4 appear to be minimal in all measured tissues, based either on low intrinsic enzymatic activity (EH4) or low absolute tissue content of the enzyme protein (EH3). Authors conclude that mEH activity could compromise effect of pharmacological blockade directed at sEH in “certain organs” and that EH3 does not appear to possess significant physiological role in EpFA metabolism.

Study design is reasonable and well executed. Approach was excellent, using what appears to be state-of-art methods and reported the enzymes in molar values rather than just mg which would otherwise have been problematic if molecular weight of one enzyme system is noticeably different from other(s). Conclusions should be amended, however, to include that EH4 as well as EH3 does not seem to play a role in EpFA turnover based upon the authors own dataset.  Also, while there is nuance that seems to support the potential  backup role of mEH in reducing the effectiveness of, sEH blockade in liver, lung and adrenal; that this may not really have the effect expected based on the lack of difference between double blockade of mEH and sEH versus sEH blockade alone. The near equivalence of mEH and sEH content in heart should be mentioned, as heart is a likely future target of sEH blockers.  It would also be helpful of other regional brain tissue (e.g., cortex) could be included, although its absence should not detract from impact of present manuscript. Might mention that importance of a redundant residual (mEH) system is similar to effect of insulin of different organ systems’ ability to clear potassium from the circulation.  Lastly, authors should emphasize more in their discussion the importance of ultrastructure in cellular enzymatic system function. Experiments on whole isolated tissues (either perfused or in vitro) or even cultured cells are indicated as a bridge between the observations in tissue homogenates and translation to the in vivo animal.   

Author Response

Comments and Suggestions for Authors

This is an in vitro study of four epoxide hydrolase enzyme systems which involved measurement of hydrolytic activity, substrate selectivity and pharmacologic selective inhibition. Study used both test systems of purified aqueous enzyme, and their quantitation in 15 human tissue homogenates. Goal was to characterize and quantitate activity of soluble epoxide hydrolase (sEH), microsomal epoxide hydrolase (mEH), epoxide hydrolase 3 (EH3) and epoxide hydrolase 4 (EH4). The overarching purpose was to assess potential theoretical contribution of each of these four EH systems to turnover of epoxy-fatty acids (EpFAs) in human tissues.  Human tissue extracts were either obtained from single patient providing broad spectrum of tissues or from pooled specimens (S-9) derived from multiple patients, both with unknown comorbidity.

Results show that sEH has favorable biochemical enzymatic characteristics for physiological clearance of EpFAs, and that mEH also could play a role in some tissues.  Potential backup role of MEH is supported by its much larger concentration (>10-fold) than sEH in liver, lung and adrenal tissue in particular. MEH was also substantially higher (2-10x) concentration than sEH in kidney, hippocampus, pancreas and stomach.  Other regions of brain were not reported.  Authors did not report whether there was any association of tissue EH content among different tissues derived from same patient. Physiological roles of EH3 and EH4 appear to be minimal in all measured tissues, based either on low intrinsic enzymatic activity (EH4) or low absolute tissue content of the enzyme protein (EH3). Authors conclude that mEH activity could compromise effect of pharmacological blockade directed at sEH in “certain organs” and that EH3 does not appear to possess significant physiological role in EpFA metabolism.

Study design is reasonable and well executed. Approach was excellent, using what appears to be state-of-art methods and reported the enzymes in molar values rather than just mg which would otherwise have been problematic if molecular weight of one enzyme system is noticeably different from other(s). Conclusions should be amended, however, to include that EH4 as well as EH3 does not seem to play a role in EpFA turnover based upon the authors own dataset.  Also, while there is nuance that seems to support the potential  backup role of mEH in reducing the effectiveness of, sEH blockade in liver, lung and adrenal; that this may not really have the effect expected based on the lack of difference between double blockade of mEH and sEH versus sEH blockade alone. The near equivalence of mEH and sEH content in heart should be mentioned, as heart is a likely future target of sEH blockers.  It would also be helpful of other regional brain tissue (e.g., cortex) could be included, although its absence should not detract from impact of present manuscript. Might mention that importance of a redundant residual (mEH) system is similar to effect of insulin of different organ systems’ ability to clear potassium from the circulation.  Lastly, authors should emphasize more in their discussion the importance of ultrastructure in cellular enzymatic system function. Experiments on whole isolated tissues (either perfused or in vitro) or even cultured cells are indicated as a bridge between the observations in tissue homogenates and translation to the in vivo animal.  

Key recommendations from reviewer’s writing:

Authors did not report whether there was any association of tissue EH content among different tissues derived from same patient.

Response: on line 206, the following precision was added: “To our knowledge, none of the tissue extracts are from the same patient.”

Conclusions should be amended, however, to include that EH4 as well as EH3 does not seem to play a role in EpFA turnover based upon the authors own dataset

Response: following the reviewer suggestion, the conclusion (line 392) was amended to now read as: “In conclusion, our results show that in human tissues sEH and to lesser degree mEH are important in the metabolism of biologically active EpFAs while EH3 and EH4 do not appear to play a role”.

The near equivalence of mEH and sEH content in heart should be mentioned, as heart is a likely future target of sEH blockers. 

Response: following the reviewer suggestion, on line 231, the subsequent sentence was added: “In the heart, a possible target of sEH inhibition [5], the concentrations of sEH and mEH are remarkably similar”.

Lastly, authors should emphasize more in their discussion the importance of ultrastructure in cellular enzymatic system function. Experiments on whole isolated tissues (either perfused or in vitro) or even cultured cells are indicated as a bridge between the observations in tissue homogenates and translation to the in vivo animal.

Response: following the reviewer suggestion, on line 366, the subsequent sentences were added: “A limitation of our approach of using whole tissue and S9 homogenates is that it cannot capture the highly localized nature of enzymatic function. Expression and function of sEH and mEH are known to be cell-type specific within an organism. For example, knocking out sEH expression specifically in podocytes is sufficient for reducing diabetic injury in the whole kidney [41]. Compartmentalization within specific organelles may similarly be important to EH function. For example, the peroxisomal sEH has been suggested to have a different role than the cytosolic sEH [42]. Future experiments on cultured cells or whole isolated tissues, where cell and tissue structure are preserved, may serve as a bridge between the in vitro observations in tissue homogenates and translation to human subjects.”